# Dynamic Changes in the Global Transcriptome of Postnatal Skeletal Muscle in Different Sheep

**DOI:** 10.3390/genes14061298

**Published:** 2023-06-20

**Authors:** Yue Ai, Yaning Zhu, Linli Wang, Xiaosheng Zhang, Jinlong Zhang, Xianlei Long, Qingyi Gu, Hongbing Han

**Affiliations:** 1Beijing Key Laboratory of Animal Genetic Improvement, College of Animal Science and Technology, China Agricultural University, Beijing 100193, China; 15731206486@163.com (Y.A.); 13121220213@163.com (Y.Z.); 15870619927@163.com (L.W.); 2Key Laboratory of Animal Genetics, Breeding and Reproduction of the Ministry of Agriculture and Rural Affairs, College of Animal Science and Technology, China Agricultural University, Beijing 100193, China; 3Tianjin Key Laboratory of Animal Molecular Breeding and Biotechnology, Tianjin 301700, China; zhangxs0221@126.com (X.Z.); jlzhang1010@163.com (J.Z.); 4Institute of Automation, Chinese Academy of Sciences, Beijing 100190, China; longxianlei2017@ia.ac.cn (X.L.); qingyi.gu@ia.ac.cn (Q.G.)

**Keywords:** skeletal muscle, transcriptome, gene expression pattern, sheep

## Abstract

Sheep growth performance, mainly skeletal muscle growth, provides direct economic benefits to the animal husbandry industry. However, the underlying genetic mechanisms of different breeds remain unclear. We found that the cross-sectional area (CSA) of skeletal muscle in Dorper (D) and binary cross-breeding (HD) was higher than that in Hu sheep (H) from 3 months to 12 months after birth. The transcriptomic analysis of 42 quadriceps femoris samples showed that a total of 5053 differential expression genes (DEGs) were identified. The differences in the global gene expression patterns, the dynamic transcriptome of skeletal muscle development, and the transcriptome of the transformation of fast and slow muscles were explored using weighted correlation network analysis (WGCNA) and allele-specific expression analysis. Moreover, the gene expression patterns of HD were more similar to D rather than H from 3 months to 12 months, which might be the reason for the difference in muscle growth in the three breeds. Additionally, several genes (*GNB2L1*, *RPL15*, *DVL1*, *FBXO31*, etc.) were identified as candidates related to skeletal muscle growth. These results should serve as an important resource revealing the molecular basis of muscle growth and development in sheep.

## 1. Introduction

Sheep (*Ovis aries*) are an important part of global livestock. Mutton is a vital source of protein for humans, with characteristics of high protein, low fat, and low cholesterol. Hu sheep, a higher prolificacy sheep breed, is native to the Taihu area in the Jiangsu and Zhejiang provinces of China. Although Hu sheep meat has the advantages of fresh tenderness, low-goaty flavor, and low fat, the carcass ratio is low, and individual growth and development are slow. Dorper is a South African sheep mutton breed with good muscle conformation, desirable carcass, and a high lean meat percentage. They have a flat back and wide waist with short and thin legs [1,2]. The binary cross-breeding with the Hu sheep as the female parent and Dorper as the male parent not only greatly increases the growth rate and meat productivity of the offspring lambs, but also overcomes the shortcomings of Hu sheep with low adult weight and less meat quantity. Improving meat quality and quantity is one of the important tasks of animal husbandry. Usually, these are determined by genetic nutritional, and environmental factors. The growth and development of skeletal muscle are closely linked to the quality and quantity of meat. Therefore, it is necessary to study the development of skeletal muscle for improving mutton production.

The skeletal muscle is the biggest organ in the body, accounting for up to 40–50% of the total body mass, and it is also the most important metabolic organ. Muscle fiber, the major component of the skeletal muscle tissue, is a syncytium that arises through the fusion of hundreds or thousands of mononucleated myoblasts during skeletal muscle development. The formation process of muscle fiber is termed myogenesis, which occurs in the stages of prenatal growth [3,4]. The number of muscle fibers is barely changed after birth [5]. Postnatal muscle growth and adaptation are mainly implemented by the remodeling of existent fibers. The growth changes in skeletal muscle after birth are involved in transverse hypertrophy and longitudinal elongation of the muscle fibers [6,7,8], regeneration, and transformation of the muscle fiber type. The postnatal period, especially the neonatal period, is the most critical stage of skeletal muscle growth and development. In humans, the growth of an infant in the first 1000 days of life is considered vital for maximizing cognitive and behavioral function, as well as minimizing the risk of chronic diseases later in life [9]. The rate of neonate growth is higher than in any other stage of postnatal development, and the majority of this growth, measured in mass, is derived from skeletal muscle [10]. As mentioned above, skeletal muscle tissue undergoes dynamic remodeling after birth to transition to the functional requirements of adult tissue.

RNA-Seq is a useful tool to measure the transcription of genes and to easily understand the physiology behind specific phenotypes [11]. It has been widely used in studies on livestock and poultry transcriptomes [12]. Analyzing the transcriptome of muscle tissue would identify more candidate genes, regulatory networks, and signaling pathways at a transcriptional level. The fetal stage is extremely important for skeletal muscle development [13]. Previous studies on skeletal muscle have mainly focused on the embryonic period. The mRNA transcriptome profiles of skeletal muscle tissue have been compared during the prenatal development of skeletal muscle in two pig breeds [14]. Dynamic transcript in the fetal development stages has been reported in goat longissimus dorsi muscle [15]. In recent years, some studies have also described the changes in transcripts of skeletal muscle growth and development after birth. One study has highlighted the possible genetic mechanisms of postnatal muscle growth and development in pigs through transcriptomic analysis of skeletal muscle in Tibetan piglets at four stages, namely 0, 14, 30, and 60 days of age [16]. There have been many studies on the muscle development of poultry embryos and post-hatching periods [17,18]. Previous studies have demonstrated differences in gene expression in sheep longissimus thoracic muscles during different stages [19], and some research has shown breed-specific expression profiles in sheep [20]. However, there is no comprehensive analysis of the dynamic changes of transcripts in skeletal muscle growth and development after birth and the differences between breeds. In this study, we selected six Hu (H), three Dorper (D), and five binary cross-breedings (HD) sheep to survey the differences in muscle fiber cross-sectional area (CSA) and transcription. We continuously sampled four stages (3 days, 3 months, 6 months, and 12 months) after birth from the quadriceps femoris of the 14 sheep to study the regulatory mechanisms and expand the molecular genetics of postnatal muscle fiber development.

## 2. Materials and Methods

### 2.1. Animals and Muscle Histology

The six H (three males and three females), five HD (three males and two females), and the three D (two males and one female) used for histological staining and RNA-seq were raised in Tianjin Farm, under the same dietary and drinking water standards. At the four developmental stages (D3 (3 days), M3 (3 months), M6 (6 months), and M12 (12 months)), the quadriceps femoris tissues of the 14 sheep were obtained by local operation. To reduce the damage and avoid the impact of surgery, the left leg was operated at D3 and M6, while the right leg was handled at M3 and M12. The skeletal muscle tissues of 14 sheep were frozen and stored in liquid nitrogen until sectioning. A total of 6 µm sections were cut using a cryostat microtome and affixed to slides. Fluorescent-conjugated (AlexaFluor488) Wheat Germ Agglutinin (WGA) (W11261, Invitrogen, Carlsbad, CA, USA) was used to stain the extracellular matrix for the calculation of the fiber cross-sectional area. The working concentration of WGA was 50 ng/mL, with incubation at 37 °C for 30 min, followed by washing in PBS three times, 5 min each time, after which the WGA staining was completed. Identification of fast and slow-twitch muscle types using ATPase staining [21].

### 2.2. Measurement of CSA and Count of Slow-Twitch Muscle Proportion

Original images were captured at 20X objective magnification using an ECHO microscope (American, RVL-100-G). We used Python 3.7 [22] and OpenCV 4.0 (https://docs.opencv.org/4.5.3/index.html (accessed on 19 August 2020)) and the work of [23] to compute the CSA of the muscle fibers. Firstly, the captured RGB image was preprocessed by gray transformation. Then, the noise and background were filtered out in the gray image using a Gaussian filter and image binarization. Subsequently, the morphological operation was implemented to select the candidate muscle fibers. The threshold for the area ranged from 1400 pixels to 80,000 pixels, and the circularity of the selected region ranged from 0.2 to 1.0. Finally, to compute the CSA of the muscle fibers more accurately, we used morphological operation again to remove the adhered muscle fibers and filled up the remained muscle fibers. The calculation of slow-twitch muscle proportion using Image-Pro Plus 6.0 software (Version 6.0.0.260 for Windows 2000/XP Professional, Serial Number: 41M60032-00032, Copyright:1993–2006 Media Cybernetics, Inc., Rockville, MD, USA).

### 2.3. Transcriptome Sequencing (RNA-Seq)

The total RNA was extracted from the quadriceps femoris using a Tissue RNA Kit according to the manufacturer’s instructions (Omega Bio-tek, Norcross, GA, USA). The RNA concentration and integrity were measured using an Agilent 2100 Bioanalyzer and Agilent RNA 6000 Nano Kit, and the RNA purity was verified by NanoDrop 2000 microspectrophotometer (Thermofisher, ND-ONE-W (A30221)). Libraries of 14 sheep were prepared and sequenced on the Illumina platform (Illumina Inc., San Diego, CA, USA) for high-throughput sequencing with read PE150 (Frasergen Bioinformatics, Wuhan, China).

### 2.4. Statistical Analysis

The CSA, the percentage of muscle fibers greater than 14.2877, and slow-twitch muscle proportion values were reported as means ± SD (H: *n* = 6, HD: *n* = 5, D: *n* = 3). The differences in CSA and slow-twitch muscle proportion were analyzed by independent-sample *t*-test with SPSS. The differences in the percentage of muscle fibers greater than 14.2877 were analyzed by one-way ANOVA with Statistical Product and Service Solutions (SPSS, IBM” SPSS Statistics version 25). The statistical significance was defined as *p* < 0.05.

### 2.5. Read Mapping and Transcript Profiling

To obtain data from raw reads to clean reads, fastp software(v2.0.0) was used to remove low-quality bases and repeated sequences of the raw sequence [24]. The clean reads were aligned to the NCBI ovine reference genome (Oar_v4.0, GCA_000298735.2, https://www.ncbi.nlm.nih.gov/datasets/genome/GCF_000298735.2/ (accessed on 19 August 2022) using Hisat2 software (v2.0.0) [25]. Then, to improve the binary alignment and map format, we used SAMtools software (v1.9)to convert SAM to BAM files, which is the binary representation of SAM and retains the same information as SAM [26]. StringTie v2.0.1 software was used to perform a reference annotation-based transcript assembly for each sample [27]. StringTie (v2.0.1) quantitative analysis was performed on the assembled transcriptome to obtain the raw count expression. To account for variations in the sequencing depth across samples, the raw count expression was normalized using the R package DESeq2 [28]. We obtained the normalized gene expression for further analysis. The differential gene expression (DE) also was calculated for muscle tissue using R package DESeq2(*p*.adjust <= 0.05 and log2FoldChange >= 0, *p*.adjust represents the corrected *p*-value) [29].

### 2.6. Functional Enrichment Analysis

The GO enrichment analysis was conducted using the Bioconductor R library cluster Profiler, applying a hypergeometric test with false discovery rate (FDR) correction (*p*.adjusted < 0.05). The KEGG pathways were found by the Bioconductor R library clusterProfiler, applying a hypergeometric test with FDR correction (*p*.adjusted < 0.05).

### 2.7. Gene Set Variation Analysis (GSVA)

We downloaded the GO gene sets (C5) from the Molecular Signatures Database (http://software.broadinstitute.org/gsea/msigdb (accessed on 19 August 2022)). The Bioconductor R software package GSVA was used to estimate the GSVA enrichment scores (ES) of each GO in each sample based on C5 and the gene expression. We conducted a one-way ANOVA of the GO pathway with ES for three varieties (H, D, and HD). Then, Spearman coefficient correlation (SSC) was calculated between the ES of the significant GO pathways and the CSA of the muscle fibers.

### 2.8. Weighted Gene Co-Expression Network Analysis (WGCNA)

Co-expression network modules were identified using gene expression and the WGCNA package (v1.51) in R. Genes with a low coefficient of variation of gene expression among all of the sample types were discarded and the remaining 20,537 genes were used for the analysis. Then, we graphically determined the optimal soft threshold for adjacency. The co-expression modules were obtained using the automatic network construction function with default settings, and the threshold to merge similar modules was set to 0.25. A module eigengene (ME) value was calculated using the module Eigengenes function for each module. The CSA of the muscle fibers was calculated as a Pearson’s correlation (PCC) between ME values. According to PCC, the modules of interest were selected. Module eigengene-based connectivity (kME) is a quantitative factor indicating the correlation (PCC) between the individual gene expression profile and ME values. Gene significance (GS), as (the absolute value of) the correlation between the gene and the trait, is defined as the log10 transformation of the Pvalue in the linear regression between gene expression and the muscle traits. Network screening (NS) is a method for identifying the genes that have a high gene significance (GS) and the member of important modules (MM) at the same time. The hub genes in this study were defined as those with kME > 0.6, GS > 0.8, and q. Wight of NS > 0.01, within the assigned module.

### 2.9. Gene Expression Clustering Trend Analysis and Biased Gene Analysis

The core algorithm of the R package Mfuzz (v.2.5.0) is based on fuzzy c-means clustering (FCM), which was used to analyze the time trend of gene expression in transcriptome data with time series characteristics and cluster genes with similar expression patterns (https://hiplot.com.cn/basic/gene-trend (accessed on 10 September 2022)). R package preprocess Core (v.1.54.0) was used to correct the gene expression from the negative binomial distribution to normal distribution (https://bioconductor.org/packages/preprocessCore/ (accessed on 13 September 2022)). We calculated the confidence intervals of the H sheep and D sheep at the same stage for each gene using the CI function of R package Rmisc (v.1.5), based on the corrected gene expression. Then, the confidence interval was compared with the corrected expression of HD sheep using the find interval function in R. If the corrected expression of HD sheep fell outside the confidence interval, the gene had an expression bias, otherwise, there was no expression bias.

## 3. Results

### 3.1. The Global Gene Expression Patterns in Different Sheep Breeds

We evaluated the morphological differences in skeletal muscle development among Hu sheep (H), Dorper sheep (D), and binary cross-breeding sheep (HD) from postnatal 3 days to 12 months. The cross-sectional area (CSA) of the skeletal muscle tissue was almost similar in the three breeds at 3 days. However, the growth of skeletal muscle developed quickly from 3 months to 12 months, and the CSA of the mean muscle fibers in the HD and D breeds was significantly greater than that in the H breed (Figure 1A,B). There was no significant difference in the CSA of the muscle fibers between the HD and D sheep, except at 6 months (Figure 1A,B). This indicates that the muscle growth ability of HD sheep and D sheep is significantly higher than that of H sheep during postnatal development. The skeletal muscle growth ability of HD sheep was greatly improved compared with H. To further explore the reasons for the differences in muscle fiber growth among breeds, the transcriptome of the skeletal muscle of the three breeds was systematically analyzed by RNA sequencing. As there was no difference in the phenotype of the three breeds at 3 days, we mainly analyzed the gene expression from 3 months to 12 months. As a result of the pairwise comparisons among the three breeds at the three developmental stages, 5053 differential expression genes (DEGs) were detected (Figure 1C and Appendix A). A total of 1015, 1790, and 1000 DEGs were found at three months from H vs. HD, H vs. D, and D vs. HD, respectively. In addition, 172, 184, and 49 DEGs were detected at 6 months, and there were 376, 453, and 14 DEGs at 12 months from H vs. HD, H vs. D, and D vs. HD, respectively (Figure 1C and Appendix A). The number of DEGs at 3 days was 254, 63, and 20 from H vs. HD, H vs. D, and D vs. HD, respectively (Appendix A). The results show that from 3 months to 12 months, the number of DEGs between D vs. HD was always lower than that between H vs. D and H vs. HD, which seems to be consistent with the difference in the CSA of muscle fibers. Next, GO enrichment items of DEGs were found using gene set enrichment analysis (GSVA), and the correlation between GO items and CSA was conducted. The greatest number of GO items were correlated highly with CSA at 3 months and 6 months in the three breeds, but not at 12 months (Figure 1D and Appendix A). The GO items highly related to CSA were significantly different among the different breeds at the different developmental stages. Therefore, the results show that there were significant differences in gene expression among the different breeds. For example, apoptotic signaling pathway and carbohydrate transmembrane transport were highly correlated with the CSA of H and HD, and had a low correlation with the CSA of D at 3 months and 6 months; mRNA metabolic process and mRNA processing were highly correlated with the CSA of D and had a low correlation with CSA of H at 3 months and 6 months. (Figure 1D and Appendix A). Moreover, the biological processes highly related to CSA were different in the different developmental stages. The RNA splicing was involved mainly in H and HD sheep at the 3 months and 6 months stages, but not at 12 months (Figure 1D and Appendix A). The pyruvate metabolic and glucose metabolic processes were associated with CAS at 6 months and 12 months in D but were involved in H and HD sheep at 3 months and 6 months (Figure 1D and Appendix A).

Then, in order to further explore the specificity of DEGs between breeds, we analyzed the stage-specific DEGs in the three breeds (Appendix A). Here, 948, 33, and 7 DEGs were found in D vs. HD at 3 months, 6 months, and 12 months, respectively. There were 957, 128, and 333 DEGs in H vs. HD at 3 months, 6 months, and 12 months, respectively. Here, 1698, 132, and 353 DEGs were observed in H vs. D at 3 months, 6 months, and 12 months, respectively (Figure 1E). The number of stage-specific DEGs also further indicates that there were significant differences in gene expression between H vs. HD and H vs. D, while the difference between HD and D was slight. Therefore, these genes may be the main reason for the difference in breeds. The KEGG analysis of the stage-specific DEGs revealed that the ECM-receptor interaction was enriched in H vs. HD at 3 months, and ubiquitin-mediated proteolysis was enriched in D vs. HD at 3 months (Figure 1E and Appendix A). The FOXO signaling pathway, citrate cycle, and PPAR signaling pathway were enriched in H vs. D at 3 months, and the PI3K-Akt signaling pathway and cell adhesion molecules were found in H vs. HD at 12 months (Figure 1E and Appendix A). The GO analysis of the stage-specific DEGs indicates biological processes, including ubiquitin-dependent, focal adhesion, and muscle atrophy at different developmental stages (Appendix A). The differential genes between HD or D and H were mainly enriched in the PI3K-Akt signaling pathway and the FOXO signaling pathway, which are important pathways for muscle hypertrophy and muscle atrophy.

### 3.2. The Dynamic Transcriptome of Skeletal Muscle Development in Different Sheep Breeds

WGCNA can be used for finding clusters (modules) of highly correlated genes and relating the modules to one another and to external sample traits [30]. We examined the correlations between CSA and gene expression data using WGCNA. The results showed that all of the expressed genes were segmented into 45 modules and the genes in the MEgreen and MEblack modules were highly correlated with CSA (Figure 2A and Appendix A). Subsequently, in order to further explore the differences in the genes in the two modules in the three breeds of sheep, these genes were divided into six groups based on similar expression patterns in H, HD, and D (Figure 2B). We found that the genes negatively correlated with the CSA of H sheep were mainly negatively correlated in clusters 1, 3, and 6 of HD and negatively correlated in clusters 1, 2, and 6 of D (Figure 2C). At the same time, the genes that were positively correlated with CSA of H were mainly positively correlated in clusters 2, 4, and 5 of HD and positively correlated in clusters 3, 4, and 5 of D, respectively (Figure 2D). The GO analysis suggests that the biological processes for maintaining normal cell activities such as RNA polymerase II basal transcription factor binding, basal transcription machinery binding, translation regulator activity, and ribonucleoprotein complex were enriched (Appendix A and Appendix A). This part of the gene was consistent with the correlation of CSA in the three breeds of sheep, so the follow-up analysis was not focused on.

We focused on the remaining 48 genes whose correlation with CSA in H was inconsistent with HD or D (Figure 2C,D), and investigated the expression of the 48 genes in H, HD, and D, respectively (Figure 2E). The results show that the expression of the 48 genes in H was significantly different from that in HD and D (Figure 2E). We speculated that 48 genes might be the cause of the difference in CSA development between breeds. For example, the expression of SYNE2, PBLD, and MAPKAPK5 in HD and D was higher than that in H at 6 months and 12 months, while the transcriptional level of SLC2A4RG, NQO2, and KARS was significantly decreased in HD and D at 3 months compared with the H sheep (Figure 2E). Because these genes were highly correlated with CSA, they might be important candidate genes for regulating muscle fiber development. To identify highly connected genes from the 48 genes, a protein network analysis was constructed. The genes including *GNB2L1*, *EEF2*, and *RPL15* were the most interactive hub genes identified in the network (Figure 2F). To facilitate the biological interpretation of 48 genes, functional enrichment analysis was performed. Figure 2G and Appendix A show that calmodulin-dependent protein kinase activity, cell–substrate junction, and focal adhesion were significantly enriched. The results show that calmodulin-dependent protein kinase activity, cell–substrate junction, and focal adhesion biological processes were significantly different in the skeletal muscle growth and development of the three breeds of sheep after birth.

### 3.3. The Global Transcriptomic Analysis of Muscle Fiber Hypertrophy

Protein synthesis is a major pathway for muscle growth after birth, resulting in an increased muscle cross-sectional area. According to the grade for the size of the cross-sectional area (CSA) of muscle fibers, the ratio of muscle fibers greater than 14.2877 (corrected) was an indicator of muscle hypertrophy. Although the average muscle fibers CSA of H from 3 months to 6 months did not increase significantly (Figure 1B), the percentage of muscle fibers greater than 14.2877 from the four stages (3 days–12 months) was increased continuously within 12 months after birth in H (Figure 3A). The results show that from 3 months to 6 months, some muscle fibers of H underwent hypertrophic growth. Subsequently, we examined the correlations between the percentage of muscle fibers greater than 14.2877 and the gene expression data using WGCNA. The results show that the gene clusters of 26 modules were correlated with the percentage of muscle fibers greater than 14.2877, and the gene cluster of the MEgreen module was highly correlated with the percentage of muscle fibers greater than 14.2877 (Figure 3B and Appendix A). To further screen the genes associated with muscle hypertrophy, 102 genes were obtained from the overlap of 1258 DEGs identified for any pair of three stages (3 months, 6 months, and 12 months) in H and hub genes in the MEgreen module (Appendix A). Subsequently, the protein network (102 genes) was constructed. The key genes including *PRKCE*, *JAK*, and *DVL1* appeared in the network center (Figure 3C). GO enrichment analysis of the 102 genes showed that the calcium-dependent protein kinase C activity and ras guanyl-nucleotide exchange factor activity were enriched (Figure 3D and Appendix A). Previous studies have shown that *PRKCE*, *JAK*, and *DVL1* played an important role in regulating myogenic differentiation and muscle development [31,32,33]. In summary, it is reasonable to use the method to screen genes related to hypertrophy. Next, we further explored the expression of genes related to hypertrophy in three sheep breeds. We identified 2292 DEGs from H and D in three stages (3 months–12 months). The DEGs related to muscle hypertrophy were obtained from the intersection of 2292 DEGs and MEgreen modular genes. As Appendix A shows, we obtained 198 genes associated with muscle hypertrophy. Then, the confidence interval for the expression of these genes was calculated at the same stage in H and D. The confidence interval was compared with the expression of HD sheep. At all stages, there were 110 genes whose expression levels were outside the confidence interval (Figure 3E and Appendix A), causing bias, and 88 genes were within the confidence interval. So, we conducted in-depth research on these 110 genes and found that the expression levels of 110 genes were close to D or H. Most of the 110 genes tended to be expressed in D (Figure 3E and Appendix A). The GO enrichment analysis of 110 genes shows the biological processes including myosin filament, muscle structure development, and contractile fiber were significantly enriched (Figure 3F and Appendix A). The biological processes, including myosin filament, muscle structure development, and contractile fiber, are involved in the growth and development of skeletal muscle after birth, which is the reason the skeletal muscle growth and development ability of H is lower than that of HD and D. Similarly, the KEGG analysis showed that the 110 genes were mainly associated with focal adhesion, ECM–receptor interaction, and the PPAR signaling pathway (Appendix A). Further study found that *SYNPO* and *FBXO31* appeared in the muscle fiber structure pathway. So, *SYNPO* and *FBXO31* might be candidate functional genes related to muscle fiber hypertrophy.

### 3.4. The Changes in the Transcriptome of the Transformation of Fast and Slow Muscles

Adult skeletal muscles in mammals are composed of different fiber types that express specific isoforms of contractile proteins and metabolic enzymes [34]. Muscle fibers are classified into slow-twitch (type I) and fast-twitch (type IIa, type IIx, and type IIb fibers) based on the mode of metabolism and the expression of specific isoforms of the myosin heavy chain. The distribution of type I and type II fibers of skeletal muscles shows high plasticity and can be influenced by various factors, such as mechanical unloading or nutrition, resulting in a change in the muscle’s functional and metabolic phenotype [21,35,36]. To explore the distribution of muscle fiber types after birth, muscle fiber types were carried out by ATPase staining. The proportion of slow muscle fibers of HD decreased significantly from 3 days to 6 months after birth (Figure 4A,B). The transcriptomic analysis showed that a total of 4143 DEGs, including 1990 upregulated genes and 2153 downregulated genes, were obtained between 3 days and 6 months in the HD sheep (Figure 4C and Appendix A). Furthermore, the GO functional enrichment analysis of the DEGs explained the roles of these genes in muscle fiber type transformation. We found that these DEGs mainly enriched the AMPK signaling pathway, cardiac muscle contraction, and citrate cycle (TCA cycle) (Figure 4D and Appendix A). AMP-activated protein kinase (AMPK) is a sensor of cellular energy status that plays a central role in skeletal muscle metabolism. Other studies have shown that AMPK plays an important role in the regulation of mitochondrial oxidative phosphorylation of skeletal muscle [37]. Peroxisome proliferator-activated receptor γ coactivator 1-α (PPARGC1A) in the AMPK signaling pathway is widely involved in a series of biological processes as a transcriptional coactivator and is highly expressed in slow-twitch myofibers, which promotes mitochondrial biosynthesis and regulates skeletal muscle metabolism by mediating the flux of glycolysis and the TCA cycle [38].

## 4. Discussion

Sheep growth performance, mainly skeletal muscle growth, provides direct economic benefits to the animal husbandry industry. However, the underlying genetic mechanisms of different breeds remain unclear. The purpose of this study was to identify candidate genes associated with the growth of sheep skeletal muscle and to investigate their potential genetic mechanisms. There are differences between males and females in skeletal muscle growth, so in order to find differences in gene expression between breeds, we aimed to avoid the influence of sex as much as possible; so, we selected sheep of different sexes for the analysis and did not conduct a sex analysis. There was a significantly different meat production performance and meat quality in Hu sheep, Dorper sheep, and binary cross-breeding (Hu and Dorper sheep), respectively. Comparative transcriptome analyses of the tissues for different breeds and at different developmental stages provide valuable insights into the question of how regulatory gene networks control specific biological processes. In this study, the comparative transcriptome was applied to explore the reason for the differences in skeletal muscle development among the three breeds.

The increase in length and girth of myofibers was a main characteristic of postnatal growth in skeletal muscle. We found that there was no significant difference in CSA among the three breeds 3 days after birth, while the CSA of H was significantly lower than that of D and H from 3 months to 12 months, which shows that the growth of skeletal muscle after birth had high plasticity. This also explains the reason the meat yield of H was lower than that of D, and the meat production ability of HD was significantly higher than that of H. In other words, the period after birth is an important stage for skeletal muscle growth and development. Previous studies have proven that both transcriptional and post-transcriptional changes are highly dynamic, particularly during the first 2 weeks after birth in mice [39]. Interestingly, based on the average CSA of the muscle fibers, the skeletal muscle in H appeared to have a growth arrest period from 3 months to 6 months. This seems to be contradictory to individual growth and development. The CSA of muscle fibers in HD and D continued to increase from 3 days to 12 months, and there was no growth plateau, which was one of the obvious differences in skeletal muscle growth between breeds. Further analysis showed that the proportion of muscle fibers greater than 14.2877 of H increased continuously. Therefore, we speculated that muscle hypertrophy was relatively weak, but biological processes such as the growth of connective tissue predominated in H from 3 months to 6 months. This might be a reason that the growth of skeletal muscle of H sheep was lower than that of the D and HD sheep after birth.

The differential expression of genes related to muscle growth is the main cause of genetic variation in different sheep breeds, indicating that the regulatory mechanism of muscle growth is different. In the study, the most gene expression changes were observed during 3 months, where 1790 DEGs were identified between H and D. Secondly, the number of DEGs was also higher between H and HD. The results were consistent with the phenotype of CSA. The correlation analysis between the pathways of GSVA and CSA of muscle fiber demonstrated that the muscle cell proliferation and regulation of the muscle system process were highly correlated with the muscle fiber CSA of H at 6 months, which suggested the muscle growth of H did not stop as the average muscle fiber CSA was presented. At the same time, Figure 1D further illustrates that there were significant differences in gene expression between breeds. The correlation between biological processes, for example, cell population proliferation, intracellular receptor signaling pathway, and pyruvate metabolic process, and CSA gradually decreased, in H and HD; meanwhile, in D, it was the opposite. In addition, the biological processes were highly correlated with protein synthesis, including mRNA metabolic process and mRNA processing, and were highly correlated with the CSA of D, but were always lowly correlated with H. The analysis of stage-specific DEGs for any pair of the three breeds showed the DEGs were enriched in different pathways. For example, at 3 months, the DEGs of H and HD were mainly enriched in the ECM–receptor interaction; however, at 3 months, the DEGs of H and D were mainly enriched in the citrate cycle (TCA cycle), focal adhesion, and PPAR signaling pathway. What is more, at 12 months, the PI3K-Akt signaling pathway and cell adhesion molecules were enriched between H and HD, while osteoclast differentiation was enriched between H and D. Therefore, the differences among breeds were accompanied by the entire growth and development, and these differences were also dynamic. Mammalian skeletal muscle possesses a unique ability to regenerate, which is primarily mediated by a population of resident muscle stem cells (satellite cells) and requires a concerted response from other supporting cell populations. Immune cells play an indispensable role in muscle regeneration [40,41,42]. In order to avoid individual differences, we used the method of continuous in vivo sampling of the same batch of individuals. Three months was the first sampling in the right leg, and twelve months was the second sampling in the same position of the right leg. From 3 months to 12 months, the muscle was mature, and the muscle reparability was relatively poor. Therefore, in addition to muscle growth, there was muscle regeneration at 12 months. In Figure 1E, a large number of immune-related pathways, including Th1 and Th2 cell differentiation, Th17 cell differentiation, and Cytokine–cytokine receptor interaction, were enriched between H vs. D and H vs. HD, which may be involved in regeneration rather than muscle growth and development, because these pathways were not enriched at 3 months.

In the early postnatal period, the longitudinal growth of muscle is also active. Muscles grow longitudinally through the accretion of myoblasts. Therefore, from 3 months to 12 months, the increase in CSA by protein synthesis is the main method of muscle growth. We attempted to explore the genes related to the CSA of muscle fiber using different gene expression patterns between H and D or HD. First of all, we performed a functional analysis of genes with the same expression patterns of H and D or H and HD. This part of the genes may not be the cause of the differences between breeds. These genes were mainly enriched in the necessary biological processes for maintaining cell activities, including RNA metabolic process and RNA polymerase II basal transcription factor binding. This was consistent with our assumption. Interestingly, we obtained 48 genes, their correlation with CSA of H was opposite to that of D and HD, and their expression in three breeds of sheep was significantly different. Finally, we obtained 48 genes as candidate functional genes for phenotypic differences. Functional enrichment analysis of 48 genes showed that they were mainly enriched in biological processes such as focal adhesion. Focal adhesions are integrin-containing, multi-protein assemblies crossing the plasma membrane that connects the cellular cytoskeleton to the surrounding extracellular matrix. They play a key role in adhesion and cell signal transduction and are the main regulators of epithelial homeostasis, tissue response to injury, and tumorigenesis [43]. Focal adhesion kinase (FAK) is a multifunctional molecule with the ability to regulate muscle formation, hypertrophy, and glucose metabolism [44].

To gain rapid growth and increase muscle mass, protein deposition is more rapid in the skeletal muscle tissue than in other tissues. Moreover, the rate of protein synthesis, which is determined by the number of ribosomes in the muscle and their efficiency in translating mRNA into protein [45], is rapid during early development, but it decreases sharply with aging [46]. In this study, the proportion of muscle fibers greater than 14.2877 was used to evaluate the hypertrophy of the existing fibers. We attempted to find genes related to hypertrophy. We selected the DEGs of H and D from the MEgreen module to explore the expression patterns of these genes in HD. Most of these 110 genes tend to be expressed in D, which was consistent with the CSA phenotype. This also provides a direction for explaining the reason the HD meat production capacity was greatly improved compared with H. Meanwhile, 110 genes were functionally enriched in biological processes and pathways related to the structure and function of skeletal muscle including calcineurin-NFAT signaling cascade, Mvosin filament, and muscle organ development. Previous studies have shown the activation of Ca^2+^-calcineurin-NFAT induces the conversion of glycolytic muscle fibers into oxidative muscle fibers by promoting the oxidative metabolism probably pathway. In addition, satellite cells are essential for generating new myofibers during regeneration and the increase in new myonuclei during hypertrophy [47]. Although muscle fibers are highly differentiated and completely exit the cell cycle, it is not unexpected to screen out genes associated with proliferation and differentiation, namely *PRKCE*, *JAK*, and *DVL1*. Satellite cells also self-renew, thus maintaining a population of quiescent, undifferentiated precursors available to respond to repeated demand in healthy adult muscle [48,49,50]. In addition, studies have shown that non-myogenic cells, such as NG2+ interstitial cells [51] and fibro adipogenic progenitors (FAPs) [52] are also essential for muscle growth and homeostasis. At 3 days, muscle fibers are immature, and there is no significant difference in CSA among the three breeds (Figure 1B). However, in 3 months–12 months, CSA shows significant differences, which means the ability of their protein synthesis of the three breeds is significantly different. The number of nuclei plays a decisive role in protein synthesis [7,51,53]; therefore, the number of nuclei stored in HD and D muscle fibers may be higher than H before and early after birth. We did not measure the related phenotype of the number of nuclei in muscle fibers. This can be used as a new research point in the future.

In many mammalian species, such as rats and mice, skeletal muscles are still immature at birth, and important changes in the fiber type profile take place during the early stages of postnatal development. Two major changes occur in the fiber type and MyHC isoform composition of rat and mouse muscles during the first weeks after birth. The first is the progressive disappearance of embryonic and neonatal MyHC, which are lost more slowly in type IIa fibers [54]. The second is the upregulation and adult fast MyHC genes and the progressive accumulation of type IIa, IIx, and IIb in specific fast-fiber subpopulations [55]. The postnatal changes in the fiber-type profile can be explained by the combined action of an intrinsic genetic program and extrinsic factors. In the present study, niacin supplementation induces muscle fiber transition from type II to type I in sheep [21], rats [56], and pigs [57]. Several genes encoding proteins involved in oxidative metabolism (*SDHA*, *COX5A*, *COX6A1*, *VEGFA*, *CPT1B*, and *SLC25A20*) were upregulated in the muscles [21]. Therefore, the transformation of muscle fiber type is fundamentally controlled by genetic mechanisms. In the study, the proportion of slow muscles decreased significantly. The oxidative phosphorylation, citrate cycle (TCA cycle), glucagon signaling pathway, and AMPK signaling pathway, which are energy metabolism-related pathways, were enriched. It is worth mentioning that the composition of muscle fiber types in skeletal muscle is closely related to meat quality. The meat yield of H, HD, and D is not only significantly different but also the meat quality of H is not as good as that of HD and D. Therefore, it is also a meaningful exploration to explore the differences in muscle fiber types among breeds. The study only compares the changes in muscle fiber types during the growth and development of HD and then focuses on the expression differences of genes related to muscle fiber-type transformation among breeds.

In summary, we revealed the differences in the growth of muscle fiber among the three sheep breeds after birth. Furthermore, the differences in the dynamic transcriptome of skeletal muscle development, the global gene expression patterns, and the transcriptome of the transformation of fast and slow muscles were explored in all three sheep breeds. The differences between H, HD, and D are dynamic and not only exist in a specific growth period but throughout the whole growth and development process after birth. Moreover, we found that the gene expression patterns of HD were more similar to D, rather than H from 3 months to 12 months. This also explains the reason the meat quality of the hybrid offspring of H and D was significantly better than H. This study systematically elucidated the mechanism of skeletal muscle development in postnatal sheep, providing evidence for the theoretical guidance for the cultivation of new breeds of high-yield and high-quality mutton sheep.

## Figures and Tables

**Figure 1 genes-14-01298-f001:**
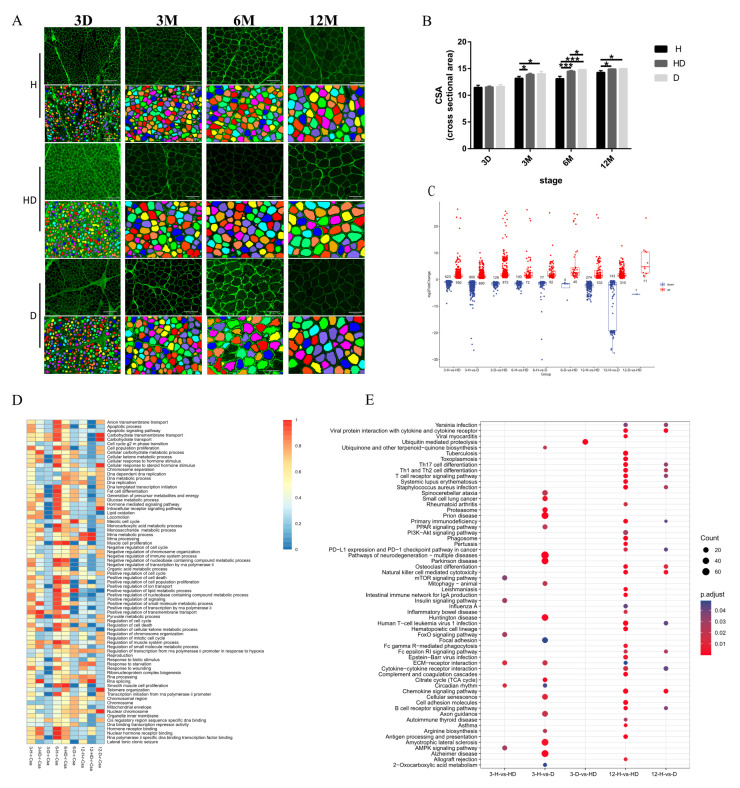
The global gene expression patterns in different sheep breeds (H: *n* = 6, HD: *n* = 5, D: *n* = 3). (**A**) The skeletal muscle tissues of the 6 H, 5 HD, and 3 D were cross-sectioned and WGA staining was performed, and histological and morphometric analyses were performed. Representative images of H, HD, and D WGA-stained muscle sections (top) and their digitally processed image (Bottom) from D3 to M12. Scale bar: 130 µm. (**B**) Quantification of average muscle fiber CSA of the three sheep breeds from D3 to M12. * *p* < 0.05, *** *p* < 0.0001. (**C**) The number of upregulated and downregulated differentially expressed genes (DEGs) in the three breeds at the three developmental stages (3 months to 12 months) through pairwise comparisons. (**D**) The significant pathways of H, HD, and D, which were enriched through GSVA analysis were associated with CSA. (**E**) KEGG pathways for (Appendix A): Stage-specific DEGs between breeds from 3 months to 12 months.

**Figure 2 genes-14-01298-f002:**
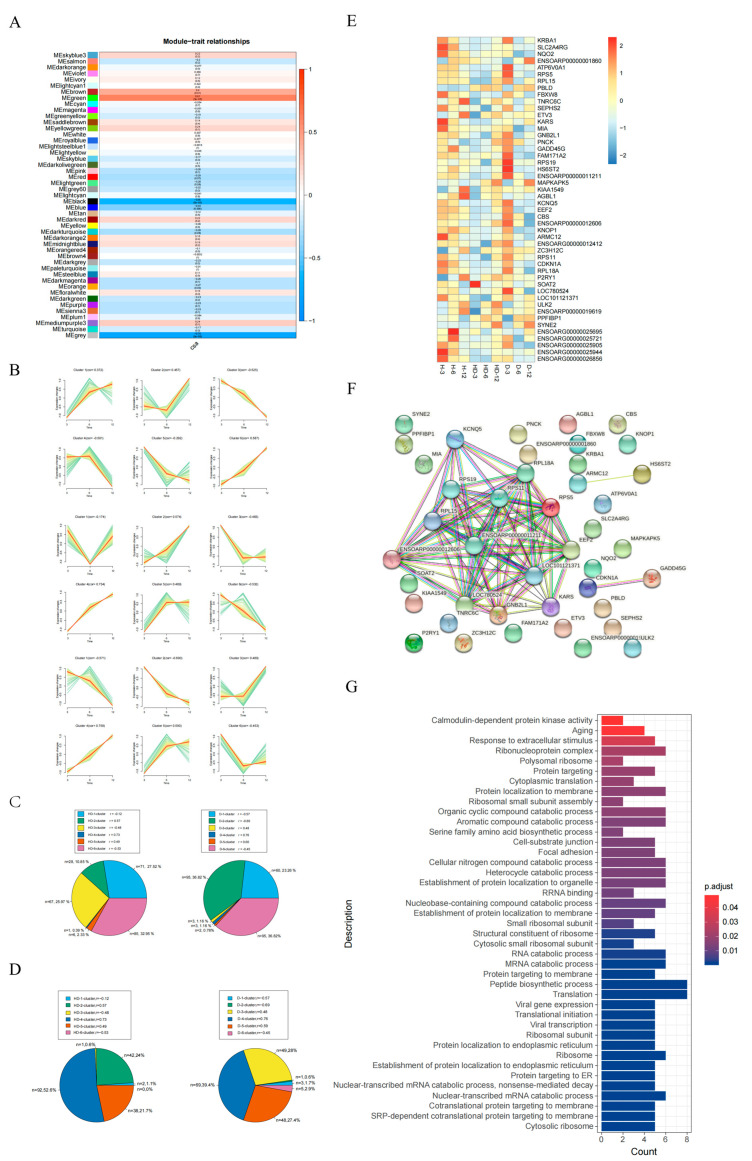
The dynamic transcriptome of skeletal muscle development in different sheep breeds. (**A**) The 45 modules (gene clusters) were obtained by gene expression and CSA correlation analysis (WGCNA analysis). (**B**) The genes of the MEgreen and MEblack modules were classified into six patterns according to their expression trends in H, HD, and D, respectively. (**C**) The genes negatively correlated with CSA in H were distributed in 6 trends in HD or D. (**D**) The genes positively correlated with CSA in H were distributed in 6 trends in HD or D. (**E**) Expression profiles of 48 genes (among the three sheep, 48 genes were inconsistent with CSA) in H, HD, and D from 3 months to 12 months. (**F**) Protein-protein interaction networks of differentially regulated genes were obtained from 48 genes. (**G**) Functional annotation cluster and gene ontology analysis of the 48 genes.

**Figure 3 genes-14-01298-f003:**
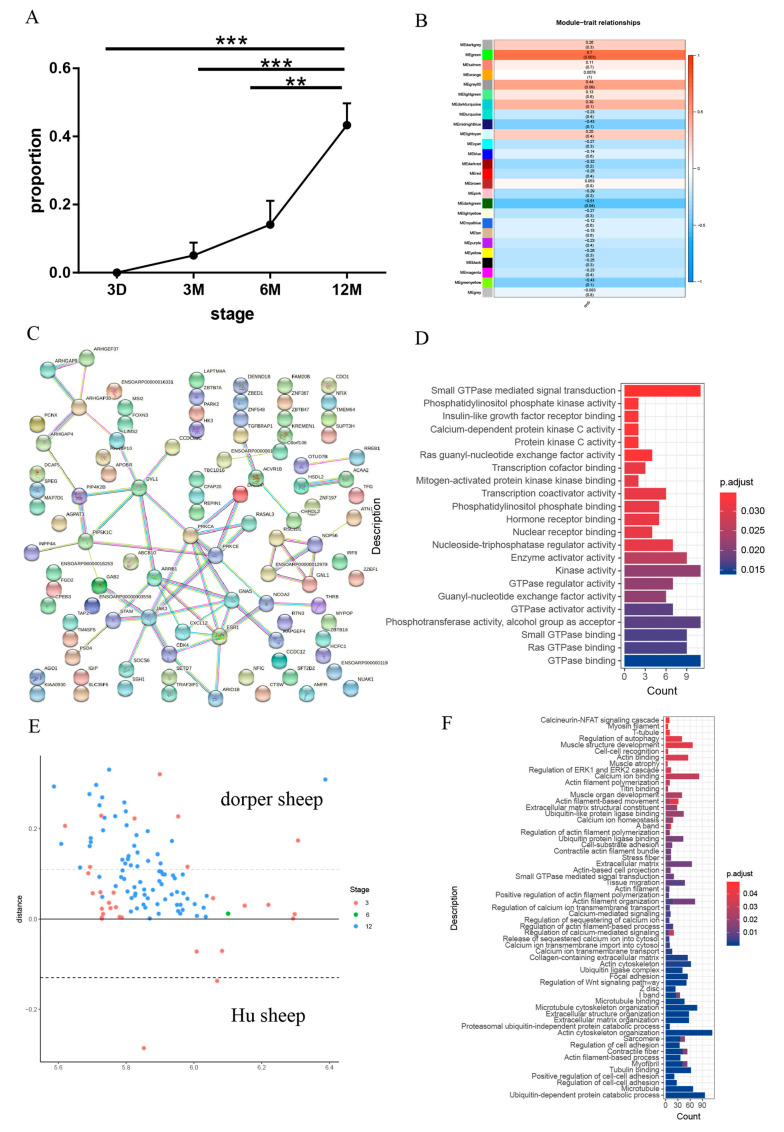
The global transcriptomic analysis of muscle fiber hypertrophy. (**A**) The changes in the proportion of muscle fibers of the CSA were greater than 14.2877 in Hu sheep from 3 months to 12 months. (**B**) The 26 modules (gene clusters) of the percentage of muscle fibers greater than 14.2877 and gene expression correlation analysis using WGCNA. (**C**) Protein-protein interaction networks of differentially regulated genes were obtained from 102 genes. (**D**) The significant GO terms of 102 intersection genes of MEgreen module hub genes and H DEGs from 3 months to 12 months. (**E**) 110 genes with biased expression. (**F**) GO items for 110 genes in (**E**). (*n* = 6, ** *p* < 0.001, *** *p* < 0.0001).

**Figure 4 genes-14-01298-f004:**
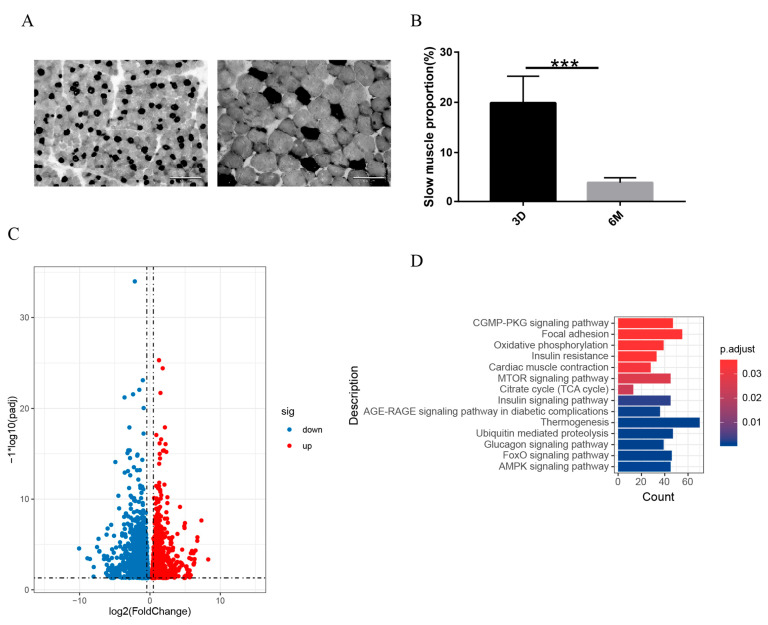
The changes in the transcriptome of the transformation of fast and slow muscles. (**A**) The distribution of fast and slow muscle fibers in HD muscle. Representative images of fiber types at D3 and M6 are shown, with the “black” areas being the slow-twitch fibers and the “grey” ones being the fast-twitch fibers. (**B**) Quantification of the proportion of the slow-twitch fibers at D3 and M6 of HD. *n* = 5, *** *p* < 0.0001. (**C**) Volcano plots for the upregulated and downregulated DEGs in the HD at 3 days and 6 months. (**D**) DEGs are significantly enriched in biological process terms.

## Data Availability

Transcriptome data were uploaded in Sequence Read Archive (SRA) database in NCBI Data Center under BioProject accession number PRJNA947918, which can be publicly accessed at https://www.ncbi.nlm.nih.gov/bioproject/?term=PRJNA947918 (Registration date: 23 March 2023).

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
