# Peer review of "Dynamic Changes in the Global Transcriptome of Postnatal Skeletal Muscle in Different Sheep"

_genes, 2023, doi:10.3390/genes14061298_

Round 1
Reviewer 1 Report
Overall, this is an interesting paper which validates the process of muscle development that has previously been studied in model organisms in a commercially important species. The representation of the results I feel needs quite a bit of editing, the authors detail much about the results for the H sheep but don’t provide much comparison between the three breeds. There is much discussion about the changes in gene expression between the time periods but not between the breeds. I believe the information is there but needs to be better presented.
While you can’t go back and re-run the same experiment with balance between the sexes there is still room to look at the analysis using sex as a co-factor. The authors did not make it known if the male sheep were intact or castrated, it’s a well-known fact that rams and wethers can have different growth patterns. It would provide an interesting comparison.
Pg1
Line 23 remove “the” at the end of the line
Line 30 italics for Latin name
Remove “The” start with “Mutton is…..”
Pg2
Line 58 change “at” to “of”
Line 74 spacing of days of age
Pg3
Line 122-123 SPSS software needs full name and reference.
Line 111-117 Only six libraries from Hu sheep were described, but the results mention the additional 8 sheep.
Pg9
Line 290 Go needs to be GO
Line 303-304 This sentence doesn’t make sense, recommend deleting
Pg11
Line 319-320 GO items for (E) gnes, are you meaning the 110 genes? This needs to be clearly stated. I would suggest re-editing all the figure captions remembering that a figure caption needs to provide enough information that the figure makes sense without reading the entire paper.
Pg12
Line 349 Quantification by which method?
Line 371 Change “were” to “had”
Overall quite good, but some editing is suggested.
Author Response
Dear Editor, Greetings!
Enclosed is the revised manuscript (ID: genes-2431986, entitled : Dynamic changes in the global transcriptome of postnatal skeletal muscle in different sheep) by Yue Ai, Yaning Zhu, Linli Wang, Xiaosheng Zhang, Jinlong Zhang, Xianlei Long, Qingyi Gu , Hongbing Han.
First of all, we would like to thank the learned editor and reviewer for valuable suggestions and comments on our manuscript. We have carefully determined the critical comments and suggestions indicated by the learned reviewer, and all suggestions and comments made of reviewer were complied with-during the revision. Besides, we have carefully examined the manuscript for minimizing the minor errors in the language expression throughout the text. As such, our manuscript has been greatly optimized.
The main modifications are as follows : 1.In the results and discussion, the description of the differences between the three breeds was added. 2. Details in the material method were supplemented. 3. References that were less relevant to this study were deleted. For convenience, all changes/editions made in the text are highlighted with a red-colored font. We now hope that our revised work will serve the purpose.
For your kind perusal, detailed point-by-point responses to the reviewer’s comments are attached to this cover letter. (Pages: 2-3).
Kind regards,
Dr. Hongbing Han
E-mail: hanhongbing@cau.edu.cn
POINT-BY-POINT RESPONSES TO THE REVIEWER’S COMMENTS
Response to the Reviewer 1 Comments
Overall, this is an interesting paper which validates the process of muscle development that has previously been studied in model organisms in a commercially important species. The representation of the results I feel needs quite a bit of editing, the authors detail much about the results for the H sheep but don’t provide much comparison between the three breeds. There is much discussion about the changes in gene expression between the time periods but not between the breeds. I believe the information is there but needs to be better presented.
Respond:We feel great thanks for your professional review work on our article. We compared the three breeds in pairs. We focused on the differences between H and HD, H and D, and described them in the article, such as Result2 and Result3. This study found that H was significantly different from D and HD in phenotype and gene expression, and the difference between D and HD was not obvious. This can be explained by the number of differentially expressed genes, the expression pattern of genes and the bias of HD gene expression. But I 'm sorry that we describe less about the results of the comparison between breeds. Now, we have added a description in the manuscript.
While you can’t go back and re-run the same experiment with balance between the sexes there is still room to look at the analysis using sex as a co-factor. The authors did not make it known if the male sheep were intact or castrated, it’s a well-known fact that rams and wethers can have different growth patterns. It would provide an interesting comparison.
Respond: Thanks for your valuable comment. In this study,the male sheep were intact. Indeed, as you emphasized, there are differences between genders, so in order to find differences in gene expression between breeds, we should avoid the influence of sexes as much as possible, so we selected sheep of different genders for analysis. There is an another important reason, the number of individual Dolper sheep is lesser, two males and one female. If gender is considered, the number of individuals is too less to conduct subsequent analysis. In future studies, we will continue to explore the effects of sexes factors on skeletal muscle growth and development
Pg1
Line 23 remove “the” at the end of the line
Line 30 italics for Latin name
Remove “The” start with “Mutton is…..”
Respond: Thank you for your careful reading of our manuscript. We have modified.
Pg2
Line 58 change “at” to “of”
Line 74 spacing of days of age
Respond: Thank you for your careful reading of our manuscript. We have modified.
Pg3
Line 122-123 SPSS software needs full name and reference.
Line 111-117 Only six libraries from Hu sheep were described, but the results mention the additional 8 sheep.
Respond: Thank you for your careful reading of our manuscript. SPSS is a data statistical analysis software. I have added the full name and version information of the software. I 'm sorry, there should be a library of 14 sheep, which has been modified.
Pg9
Line 290 Go needs to be GO
Line 303-304 This sentence doesn’t make sense, recommend deleting
Respond: Thank you for your careful reading of our manuscript. We have modified. This sentence (Line 303-304)had been deleted.
Pg11
Line 319-320 GO items for (E) gnes, are you meaning the 110 genes? This needs to be clearly stated. I would suggest re-editing all the figure captions remembering that a figure caption needs to provide enough information that the figure makes sense without reading the entire paper.
Respond: Thank you for your kind suggestions. In the manuscript, we modified the figure captions, which is not clear.
Pg12
Line 349 Quantification by which method?
Line 371 Change “were” to “had”
Respond: Thank you for your kind suggestions. In the manuscript, we modified the figure captions, which is not clear. Studies have shown that Peroxisome proliferator-activated receptor γ coactivator 1-α (PPARGC1A), was highly expressed in slow-twitch myofibers[1] (Line 349), by comparing the expression levels of PPARGC1A in pectoralis major(It is composed of fast muscle) and soleus muscle(It is composed of slow muscles), this conclusion was drawn.
- Ma M, Cai B, Kong S, Zhou Z, Zhang J, Zhang X, et al. PPARGC1A Is a Moderator of Skeletal Muscle Development Regulated by miR-193b-3p. International journal of molecular sciences. 2022;23(17).

Reviewer 2 Report
The authors describe mRNA changes in different sheep breeds regarding their skeletal muscle development after birth comparing four developmental stages (3 days, 3 months, 6 months,12months).
Introduction is focused, Materials and methods is divided accordingly, descriptions are acceptable. Results and discussion are comprehensive, discussion provides further ideas generated by the results.
Notices
Line 21: Please define what is WGCNA, like other acronyms in this part of the manuscript.
Line 101: …Echo microscope (American)…
Please give more specification.
Line 102: Please give versions of Python and OpenCV.
Line 110: Please give reference, firm, country of Image-Pro Plus 6.0 software.
Line 115: The same as above; NanoDrop 2000 (firm, country)
Line 116: Illumina platform: Please give more detail.
Line 118: You might also consider to use multivariate general linear model.
Line 142: Please define FDR.
Author Response
Cover letter
Dear Editor, Greetings!
Enclosed is the revised manuscript (ID: genes-2431986, entitled : Dynamic changes in the global transcriptome of postnatal skeletal muscle in different sheep) by Yue Ai, Yaning Zhu, Linli Wang, Xiaosheng Zhang, Jinlong Zhang, Xianlei Long, Qingyi Gu , Hongbing Han.
First of all, we would like to thank the learned editor and reviewer for valuable suggestions and comments on our manuscript. We have carefully determined the critical comments and suggestions indicated by the learned reviewer, and all suggestions and comments made of reviewer were complied with-during the revision. Besides, we have carefully examined the manuscript for minimizing the minor errors in the language expression throughout the text. As such, our manuscript has been greatly optimized.
The main modifications are as follows : 1.In the results and discussion, the description of the differences between the three breeds was added. 2. Details in the material method were supplemented. 3. References that were less relevant to this study were deleted. For convenience, all changes/editions made in the text are highlighted with a red-colored font. We now hope that our revised work will serve the purpose.
For your kind perusal, detailed point-by-point responses to the reviewer’s comments are attached to this cover letter. (Pages: 2).
Kind regards,
Dr. Hongbing Han
E-mail: hanhongbing@cau.edu.cn
POINT-BY-POINT RESPONSES TO THE REVIEWER’S COMMENTS
Response to the Reviewer 2 Comments
Line 21: Please define what is WGCNA, like other acronyms in this part of the manuscript.
Respond: Thank you for your careful reading of our manuscript. We have added the full name of WGCNA
Line 101: …Echo microscope (American)…
Please give more specification.
Line 102: Please give versions of Python and OpenCV.
Line 110: Please give reference, firm, country of Image-Pro Plus 6.0 software.
Line 115: The same as above; NanoDrop 2000 (firm, country)
Line 116: Illumina platform: Please give more detail.
Respond: Thank you for your kind suggestions. We have added detailed information about these software, instruments, etc.
Line 118: You might also consider to use multivariate general linear model.
Respond: Thank you for your kind suggestions. We will adopt your suggestions and use multivariate general linear model for data analysis. In this study, analysis of CSA differences between the two breeds at the same stage of development and slow muscle ratio differences of HD at 3D and 6M, The data are in line with the conditions of independent t test:1.Two groups of samples are independent of each other, 2.Conform to the normal distribution 3.Two groups of samples have equal variance. Therefore, these two parts of the analysis chose the independent t test. In addition, the analysis of the percentage of muscle fibers greater than 14.2877 is to compare the differences between any two time points in the three developmental stages and meet the conditions of one-way ANOVA. Therefore, one-way ANOVA was selected in this study.
Line 142: Please define FDR.
Respond: Thank you for your careful reading of our manuscript. We have added the full name of FDR (False discovery rate).
